# In Touch with the Heartbeat: Newborns’ Cardiac Sensitivity to Affective and Non-Affective Touch

**DOI:** 10.3390/ijerph18052212

**Published:** 2021-02-24

**Authors:** Letizia Della Longa, Danica Dragovic, Teresa Farroni

**Affiliations:** 1Developmental Psychology and Socialization Department, University of Padova, 35131 Padova, Italy; letizia.dellalonga@phd.unipd.it; 2Department of Paediatric Unit, Hospital “S. Polo”, 34074 Monfalcone, Italy; danica.dragovic@aas2.sanita.fvg.it

**Keywords:** newborns, affective touch, heart rate variability, autonomic self-regulation

## Abstract

The sense of touch is the first manner of contact with the external world, providing a foundation for the development of sensorimotor skills and socio-affective behaviors. In particular, affective touch is at the core of early interpersonal interactions and the developing bodily self, promoting the balance between internal physiological state and responsiveness to external environment. The aim of the present study is to investigate whether newborns are able to discriminate between affective touch and non-affective somatosensory stimulations and whether affective touch promotes a positive physiological state. We recorded full-term newborns’ (N = 30) heart rate variability (HRV)—which reflects oscillations of heart rate associated with autonomic cardio-respiratory regulation—while newborns were presented with two minutes of affective (stroking) and non-affective (tapping) touch alternated with two minutes of resting in a within-subject design. The results revealed that non-affective touch elicits a decrease in HRV, whereas affective touch does not result in a change of HRV possibly indicating maintenance of calm physiological state. Thus, newborns showed cardiac sensitivity to different types of touch, suggesting that early somatosensory stimulation represents scaffolding for development of autonomic self-regulation with important implications on infant’s ability to adaptively respond to the surrounding social and physical environment.

## 1. Introduction

The infant’s abilities to assimilate information and to adapt to the sensory environment are critical for cognitive and social development. To attend to the environment effectively, infants must be able to regulate the balance between internal physiological needs and responsiveness to the external stimulation [1]. Self-regulation refers to the mechanism that supports the integration and organization of various processes at the physiological, emotional, attentional and cognitive levels [2]. In a developmental perspective, higher order mechanisms of self-regulation (e.g., cognitive control) are not predetermined but rather emerge from the interplay between early developing lower level physiological regulatory functions and environmental influences [3]. Given the essential importance of maintain homeostasis and regulating organism-environmental exchanges in the neonatal period, physiological regulatory capacities are a primary objective since the very first stages of development and then are hierarchically integrated with emerging new skills, such as emotion and attention regulatory processes [3]. Specifically, initial regulatory capacities can be evidenced in the newborns’ ability to adaptively modulate the neurobehavioral state, meaning the basic biological rhythm that alternates sleep and wake states. Newborns spend most of their time resting in the sleeping state, alternating cycles of quiet sleep (characterized by closed eyes, behavioral quiescence, regular and deep breathing) and active sleep (characterized by closed eyes, intermittent ocular movements, isolated facial and body movements, irregular breathing) with short periods of wakefulness (characterized by open eyes, general body movements and high muscle tone, irregular heart rate and respiratory patterns) [4]. Importantly, neurobehavioral organization mediates the reactivity to external stimulation while maintaining internal stability [5], with important implications for infants’ engagement with the surrounding physical and social environment [6]. In early infancy, sleep is essential for supporting neural development and regulation of sleep-wake states has a potential impact on developmental trajectories [7]. Indeed, infants with more organized sleep-wake cycles showed better arousal regulation, attention and explorative behaviours in the first months of life [8].

The autonomic nervous system (ANS) plays a central role in coordinating physiological responses to environmental changes. The development of autonomic regulation begins in utero: indices of both sympathetic activation and vagal modulation increase with growing gestational age, as measured by changes in autonomic regulation of the cardiovascular system and neurobehavioral state (fetal activity; [9]). In particular, self-regulation skills and arousal modulation seem to be mediated by parasympathetic activity (vagal tone), which is commonly measured through modulation of heart rate variability (HRV; [10]). HRV is defined as the variation in the time interval between successive heartbeats and reflects autonomic regulation underpinning the organism’s ability to continuously adjust to internal and external challenges [11]. In this context, HRV could provide a noninvasive tool to obtain information about cardiac regulatory mechanisms and in tune a useful insight about ANS development and functioning [12]. Among the time domain methods used to assess HRV, the most common index is the root mean squared successive differences (RMSSD), which is based on the differences between adjacent heart periods and correlates with high frequency HRV. RMSSD reflects beat-to-beat variance in heart rate, filtering out lower frequency variability and capturing fluctuations associated with the respiratory cycle [13]. Thus, this index estimates the vagal influence on HRV [14], which primary consists in fostering synchronization between respiratory and cardiovascular processes during metabolic and behavioral changes [10]. Indeed, HRV is closely connected with neurobehavioral states. In particular, during quiet sleep the parasympathetic system is dominant and the respiratory drive is under autonomic control resulting in slower and more regular respiration and increased HRV [15].

During the first days of life, HRV increases, indicating maturation of cardiovascular regulation that accompanies the transition from fetal to postnatal life [16]. The neonatal period is characterized by venerable physiological changes, suggesting the essential role of cardiovascular regulatory mechanisms for early adaptation in the extrauterine environment [17]. The development of the cardiac autonomic innervation is not fully mature at birth, as suggested by the fact that newborns have high heart rate, which indicates a sympathetic predominance associated with decreased vagal activity [18]. Full-term newborns have shown a tendency to increase HRV within the first postnatal days and to modulate HRV according to neurobehavioral states, suggesting a rapid maturation of autonomic cardiac control [16]. Moreover, evidence from preterm newborns showed lower HRV compared to full-term newborns, indicating a delay of parasympathetic influence on autonomic control due to prematurity [19]. Thus, investigating newborns HRV provides a unique manner to obtain information about early development of sympathovagal balance, which indicates progressive maturation of ANS that may critically contribute to postnatal adaptation. Neonatal cardiac vagal tone has been shown to predict regulatory outcomes, as higher HRV is related to better cognitive performance, emotional regulation and social functioning [20,21]. Moreover, resting HRV seems to be associated with the ability to modulate reactions towards novel stimuli or mildly stressful situations [22] and, indeed, it has been considered a biomarker of individual differences in self-regulation [23] and susceptibility to stress [24]. Notably, the development of regulatory abilities can be modulated by environmental factors, such as early maternal care [25], indicating that regulatory capacities are open to contextual influences integrating internal physiological processes and external sensory inputs [3].

The sense of touch represents a link between the body (internal physiological states) and the external world (sensory stimulation), suggesting it may be critically involved in the development of self-regulation skills, mediated by activation of the parasympathetic system. Early affective interactions, which hinge on physical contact and somatosensory stimulation, have been shown to critically impact the functioning of ANS and stress reactivity system, contributing to self-regulatory processes across the life span [26,27] and to the formation of affiliative bonds [28]. Indeed, the impact of maternal contact on the neuroendocrine and autonomic systems seems to be specifically related to tactile stimulation, rather than food or temperature changes [29], indicating that early somatosensory experiences may critically modulate biochemical and physiological states, facilitating growth and development of functional self-regulation.

Consistent with the literature from non-human primates and rats [30,31,32], in humans close bodily contact between mother and her newborn has been shown to promote regulation of infant’s temperature, respiration and quiet sleep state since the very earliest stages of life [33,34], More specifically, early tactile interactions may play an essential role in increasing parasympathetic activity, as reflected by a decrease in heart [35] and in modulating stress reactivity, as reflected by a decrease in cortisol and increase in oxytocin levels [36]. Six-month-old infants exposed to a distress event, during which the mother suddenly posited a neutral expression interrupting social exchange (still face paradigm), showed lower cortisol levels if mothers provided tactile stimulation during the procedure [37], resembling behavioral findings that support the essential role of touch in modulating infants’ affect and attention during momentary maternal unavailability [38]. Indeed, mothers commonly use touch to regulate their infant affective state and reduce infants’ crying when they are distress [39,40], indicating that early affective interactions, mediated by physical contact, prevent the potentially adverse effects of stress on infants and promote the development of behavioral and physiological self-regulation.

In line with these results, interventions based on skin-to-skin contact (kangaroo care and massage therapy) have been shown to support more organized sleep-wake cycle and more restful sleep [41,42], reduce autonomic stress responses and promote beneficial physiological conditions in preterm infants [8,43,44]. More specifically, preterm newborns showed a decrease in HR and an increase in blood oxygenation levels during gentle stroking, but not during static touch [45], suggesting that the beneficial autonomic effects of physical contact are mediated by the quality of the tactile interaction rather than the mere presence of somatosensory input.

More specifically, gentle dynamic stroking is considered a particular type of touch, linked to a neurophysiologically specialized system of cutaneous afferents, named C-tactile system. Whereas the sensory-discriminative functions of touch are conveyed via the activation of fast conducting myelinated Aβ afferents, which project to somatosensory cortices and provide precise information about the spatio-temporal properties of mechanical stimulation [46], the affective and motivational properties of touch are conveyed by slow-conducting, unmyelinated C-tactile afferents, which follow a separate pathway projecting directly to the posterior insula and to other crucial nodes of the social-brain network, including the posterior superior temporal sulcus, medial prefrontal cortex, and dorsal anterior cingulate cortex [47,48,49]. C-tactile afferents are selectively activated by dynamic tactile stimuli delivered at slow velocity (1–10 cm/s), low force (0.3–2.5 mN) and neutral (skin-like) temperature [50,51], implying that the C-tactile system is tune to encode and process tactile stimuli that are likely to carry a social relevance [52]. Additionally, activation of C-tactile afferents positively correlates with subjective reports of pleasantness [53] and elicits implicit positive reactions [54], suggesting that the C-tactile system is linked to positive affect and to the rewarding value of social interactions. Taken together, the neurophysiological properties of affective touch point to an interoceptive function of interpersonal tactile interactions providing a link between external sensory information and internal affective states in order to support the perception of subjective feelings and to maintain the organism’s physiological balance [55]. Thus, affective touch may represent a connection between early sensory experiences and the development of behavioral and physiological self-regulation.

Developmental studies suggest that from the earliest stages of life, affective touch may have a calming and rewarding effect with a prominent role in modulating physiological arousal and in promoting social bonding [56]. Notably, the ontogeny of affective touch can be considered even prenatally. Evidence suggests that touch is the earliest sensory modality to develop in utero [57] and by 26 weeks of gestation foetuses start to actively respond to vibration applied on the mother’s abdomen with increased movements and heart rate acceleration [58]. Specifically, external touch produces vibroacustic stimulation in the amniotic fluid yielding to oscillations of lanugo hairs and stimulation of C-tactile system, which in tune lead to enhanced vagal tone and release of oxytocin and gastrointestinal hormones [59]. This points to a crucial role of tactile stimulation for foetuses grown and wellbeing, shaping brain development to respond to social rewards. During the first hours after delivery, skin-to-skin contact between mother and newborn has been shown to enhance newborns’ quiet sleep state, supporting the development of neurobehavioural regulation [33,60]. Moreover, affective touch has been shown to be effective in regulating infants’ behavioral and physiological responses to stress during periods of momentary maternal unavailability [37,38], in reinforcing infants’ social behaviors, such as eye-contact and smiling during face-to-face interactions [61] and in facilitating learning of contingent facial information [62]. However, the development of the neurophysiological mechanisms that supports tactile processing since birth is not fully understood.

We believe that investigating newborns’ physiological sensitivity to tactile stimulations since the very first hours of life would provide important evidence for the understanding of early self-regulatory skills and the cascading effects of affective tactile interactions on promoting socio-emotional and cognitive development. Given that touch is the first sensory modality to anatomically and functionally mature in fetal life [58,63] and affective tactile experiences are abundant in early postnatal development, it is reasonable to hypothesize that affective touch provides the developing brain with its primary sensory and affective templates, representing a neurophysiological underpinning of responsiveness to external sensory input and development of self-regulation abilities. These early capacities represent essential scaffolding for all interactions with the external physical and social environment, pointing to the importance of studying physiological responses to tactile stimulation in the neonatal period.

The present study aims to investigate newborns’ physiological sensitivity to different types of dynamic tactile stimulations emphasizing the importance of flexible autonomic regulation to adaptively respond to sensory input. Tactile experiences constitute a primary manner by which newborns receive information from the external environment and develop the sense of bodily self. Thus, the sense of touch represents the foundation for all physical and social interactions with the external world [64]. More specifically, the early ability to discriminate between different tactile stimulations and to adaptively modulate internal physiological state in response to specific sensory inputs may represent early scaffolding for efficient processing of sensory information and the development of autonomic self-regulation skills. 

We first hypothesized that newborns are able to discriminate between affective touch and non-affective somatosensory stimulations. More specifically, we expected that gentle stroking (affective touch) would promote vagal activity, as reflected by an increase in infants’ HRV, whereas tapping movement (non-affective touch) would be associated with vagal withdrawal, as reflected by a decrease in infants’ HRV. In addition, we were interested in investigating the modulation of newborns’ HRV during the post-stimulation resting period, in line with previous results that found the effects of affective touch to be maintained beyond the period of tactile stimulation [45]. Finally, we propose to explore possible individual differences in autonomic cardiovascular regulation. Previous findings suggest a relevant inter-subject variability in newborns’ HRV [65], pointing to the need of taking an individual-based approach. Heart rate variability is modulated by many factors that may influence baseline HRV parameters and responsiveness to external stimulation [16]. In particular, in the present study we included the variables gestational age, birth weight, type of delivery (vaginal delivery vs. caesarean section) and postnatal age, which have been found to be important determinants of HRV in newborns [16,18,65]. We hypothesized that neonatal individual differences not only are related to HRV level during resting, but can also modulate newborns’ cardiac responses (HRV changes) to different tactile stimulations.

The results of the present study would be of particular interest in order to better understand whether newborns show specific autonomic responses to different tactile stimulations in order to adjust their physiological state to the sensory input and whether affective touch may promote self-regulation abilities since birth.

## 2. Materials and Methods

### 2.1. Participants

The study was conducted at the Pediatric Unit of Monfalcone Hospital (Gorizia, Italy) where all infants were born. Newborn participants were recruited at the hospital during their and their mother’s stay. In agreement with the medical and nursing staff, the experimenter informed parents about the research topic and the whole procedure. If the parents agreed and gave written consent for their child’s precipitation, the experimenter brought the electrocardiography equipment into the hospital room and the study was conducted as soon as the newborn was resting in a sleeping state. Thirty-nine newborns took part in the study. Nine newborns were excluded from successive analyses as they were born preterm (N = 2), were too young (less then one hour of life N = 1), failed to complete the task because of fussiness (N = 1), or due to non-valid physiological data (N = 5). The final sample consisted of thirty healthy full-term newborns (13 females and 17 males; mean age 48.94 (SD = 29.35) hours at the time of test). All infants met the screening criteria for normal delivery (gestational age > 37 weeks, birth weight > 2500 g, Apgar score ≥ 8 at 5 min after birth—which is a method to summarize the health of newborn against infant mortality based on five criteria [66,67]. Moreover, given the fact that HRV changes significantly during the first 6 h of life, followed by a gradual normalization (Olivera et al., 2019), we decided to include in our experimental sample only newborns aged more then 6 h. The local Ethical Committee of Psychological Research (University of Padova) approved the study protocol.

### 2.2. Stimuli and Procedure

The study took place in the hospital rooms. In line with previous studies that investigated newborns’ HRV [15,68,69], we tested infants when they were asleep or in a calm state with closed eyes, lying supine in their cradle. Before starting the experimental session, three pediatric electrodes were placed on newborns’ chests to measure cardiac electrical activity. The experimental session lasted 10 min, alternating 2 min of resting with 2 min of tactile stimulation. We select this interval length as it has been shown that two-minute recording is sufficient for obtaining accurate measures of RMSSD [70]. Newborns were presented with two types of touch (affective touch vs. non-affective touch), performed by an experimenter trained in order to deliver the tactile stimulation with a constant velocity/rhythm and pressure. During the affective touch condition the trained experimenter gently caressed the infant’s forehead with her hand at approximately 3 cm/s, while during the non-affective touch condition the experimenter performed a rhythmic tapping with a paintbrush on the infant’s forehead. The choice of the forehead as site for the tactile stimulation was made because it is innervated by C-tactile afferents [71] and is an available and ecological tactile interaction when the newborns are lying in their cradle. The two types of touch were selected in the attempted to match the extent of sensory input by equating the contact area and the stimulation rate, while differentiating socio-affective value by varying both source of touch and spatio-temporal dynamics of the tactile interaction. Stroking by a human hand represents an optimal tactile stimulation for activating the C-tactile afferents and it carries a social and affective significance; whereas tapping with a brush should not activate the C-tactile system, suggesting a neutral and non-social meaning. Notably, a neuroimaging study found a main effect of type of touch indicating a greater activation in the posterior insula, which is a key region in determining the valence of touch stimulation, during stroking compared to tapping [72]. Moreover, such difference was amplified when participants were touched with a hand rather than velvet stick, indicating that the combination of perceptual differences and affective factors related to direct skin-to-skin contact modulates the neural responses to different types of touch [72]. Based on these findings, in the present study we propose to ensure ecological validity manipulating both spatio-temporal characteristics and direct interpersonal contact of affective vs. non-affective touch. The order of the tactile conditions was counterbalanced between participants, resulting in 15 infants perceiving the affective touch first and 15 infants perceiving the non-affective touch first (Figure 1).

### 2.3. Electrophysiological Data Recording and Processing

Electrocardiogram (ECG) was recorded by means of three Ag/AgCl electrodes placed on the newborns’ chests using a multimodality physiological monitoring device that encodes biological signals in real-time (ProComp Infiniti; Thought Technology, Montreal, QC, Canada) which is a computerized recording system approved by the U.S. Food and Drug Administration (FDA). ECG signal was recorded continuously via a 12-bit analogue-to-digital converter with a sampling rate of 256 Hz and stored sequentially for analysis. First, the ECG signal was visually inspected, and artifacts were corrected by means of a piecewise cubic splines interpolation method that generates missing or corrupted values into the IBIs series. Then, heart rate variability was calculated for each experimental period considering the squared root of the mean squared differences between successive heart periods (RMSSD). This value is obtained by first calculating each successive time difference between heartbeats in milliseconds, which are then squared; the result is averaged and finally the squared root of the total is obtained [11]. This index reflects the beat-to-beat variance in HR (short-term HRV) and it is a primary time-domain measure to estimate vagal influence on HRV [14].

### 2.4. Statistical Analysis

All statistical analyses were performed using R, a software environment for statistical computing and graphics [73]. In order to investigate whether newborns show physiological sensitivity to different tactile stimulations we compared changes of HRV during affective vs. non-affective tactile stimulation. More specifically, first we calculated a differential RMSSD score subtracting the RMSSD value measured during the pre-stimulus period from the RMSSD value measured during the touch period on a participant-by-participant basis, to the end of considering each individual baseline level. Then, we performed a paired *t*-test to compare the variation of RMSSD in response to the two types of touch and simple *t*-test comparing the differential RMSSD score with the null level (zero). To the end of investigate whether the effects of touch extend beyond the period of actual stimulation, we performed the same analyses on the resting period following the tactile stimulation.

Moreover, to analyze newborns’ modulation of HRV, we carry out mixed models using “lmer” from the “lme4” package [74]. In order to compute R-squared for the models, we used “r.squaredGLMM” from MuMIn package [75], which takes into account the marginal R-squared (associated with fixed effects) and the conditional one (associated with fixed effects plus random effects). For each model, we reported the marginal R-squared. The *p*-value was also calculated using the “lmerTest” package [76]. The choice of using a mixed-effects model approach was determined by the possibility to take into account fixed effects, which are parameters associated with an entire population as they are directly controlled by the researcher, and random effects, which are associated with individual experimental units randomly drawn from population [77,78]. Akaike information criterion (AIC) model comparison has been used to compare a set of models fitted to the same data [79,80]. The model that produces the lowest AIC value is the most plausible [81].

Specifically, six nested mixed-effects models were tested. In each model, RMSSD score was the dependent variable. The null model (Model 0) included only the random effect of Participants; the first (Model 1) included Presence of touch (2 levels; resting period vs. tactile stimulation) as fixed factor and Participants as random factor; the second (Model 2) included also the Type of touch (2 levels; affective touch vs. non-affective touch) as a fixed factor; the third (Model 3) added the interaction between Presence of touch and Type of touch to Model 2. Moreover, we wanted to control whether the order of tactile stimulation (two levels: first vs. second stimulation) may have influenced the heart rate variability and whether there was an interaction effect between the experienced type of touch and the order of tactile stimulations (experiencing affective touch before the non-affective touch and vice-versa). Therefore, we tested two additional models including also the order of tactile stimulations as a fixed factor (Model 4) and the interaction (Model 5).

In addition, we were interested in investigating whether the effects of touch on infants’ HRV lasted over the time of stimulation. To this end, we considered the RMSSD index during the resting period following the tactile stimulation, including a three-points time dimension (pre-stimulation, stimulation and post-stimulation). Therefore, we perform a second model comparison including six nested mixed-effects models. Again, in each model, RMSSD score was the dependent variable. The null model (Model 0) included only the random effect of Participants; the first (Model 1) included the Time dimension (3 levels; pre-stimulation vs. tactile stimulation vs. post-stimulation) as fixed factor and Participants as random factor; the second (Model 2) included also the Type of touch (2 levels; affective touch vs. non-affective touch) as a fixed factor; the third (Model 3) added the interaction between Time and Type of touch to touch. We tested two additional models including the order of tactile stimulations (2 levels: first vs. second stimulation) as a fixed factor (Model 4) and the interaction (Model 5).

Finally, in order to explore the relationship between neonatal individual differences (in terms of gestational age, birth weight, type of delivery and postnatal age) and modulation of HRV, we carried out correlations using the “cor.test” function, which returns both the correlation coefficient and the significance level of the association between paired samples.

## 3. Results

To the end of investigating whether newborns show physiological sensitivity to different types of tactile stimulation, as reflected by HRV modulation, we considered the differential RMSSD score during the tactile stimulation compared to the pre-stimulus resting period. A paired *t*-test was performed comparing the variation of RMSSD in response to the two types of touch (affective vs. non-affective). The results revealed a different modulation of the HRV based on the type of touch administered [t (29) = 2.71, *p* = 0.011, Cohen’s d = 0.66]. In addition, two simple *t*-test comparing the differential RMSSD score with the null level (zero) were separately performed for each type of tactile stimulation in order to explore whether the fact of being touch elicited a change in newborns’ HRV compared to the pre-stimulus resting period. The results revealed that only when newborns’ were stimulated with non-affective touch, they showed a lower HRV, as indexed by a decrease of RMSSD [t (29) = −2.81, *p* = 0.009, Cohen’s d = −0.51; *p*-value adjusted for multiple comparisons using Bonferroni correction, *p* < 0.025]. On the contrary, when infants were stimulated with affective touch, they did not show a modulation of HRV compared to the pre-stimulus resting period [t (29) = 0.41, *p* = 0.682, Cohen’s d = 0.07; *p*-value adjusted for multiple comparisons using Bonferroni correction, *p* < 0.025] (Table 1; Figure 2).

Then, we considered the variation in the RMSSD index during the resting period following the tactile stimulation in the two experimental conditions (affective touch vs. non-affective touch). Specifically, we calculated the differential RMSSD score during the post-stimulation resting period compared to the pre-stimulation resting period for both tactile conditions. A paired *t*-test was performed comparing the variation of RMSSD following the two types of touch (affective vs. non-affective). In line with the results found during the tactile stimulation, the different modulation of the HRV based on the type of touch administered tends to be maintained beyond the period of tactile stimulation [t (29) = 2.04, *p* = 0.051, Cohen’s d = 0.62]; although statistical significance was not reached. In addition, two simple *t*-test comparing the differential RMSSD score with the null level (zero) revealed that only when newborns’ were previously stimulated with non-affective touch, they showed a lower HRV, as indexed by a decrease of RMSSD [t (29) = −2.37, *p* = 0.025, Cohen’s d = −0.43; *p*-value adjusted for multiple comparisons using Bonferroni correction, *p*< 0.025]. On the contrary, when infants were stimulated with affective touch, they did not show a modulation of HRV compared to the pre-stimulus resting period [t (29) = 1.13, *p* = 0.267, Cohen’s d = 0.21; *p*-value adjusted for multiple comparisons using Bonferroni correction, *p* < 0.025] (Table 2).

In light of these results, we further analyzed infants’ changes in HRV by using a model comparison approach. Specifically, we compared six nested mixed-effects models. In each model, RMSSD score was the dependent variable (see Table 3).

The likelihood ratio test showed that Model 3 was the best at predicting the collected data. The main effect of Type of touch and the interaction effect between Presence of touch and Type of touch emerged (Table 4), suggesting that newborns’ HRV is differently modulated based on the type of tactile stimulation (Figure 3).

In addition, we were interested in investigating whether the effects of touch on infants’ HRV lasted over the time of stimulation. To this end, we considered the RMSSD index during the resting period following the tactile stimulation, including a three-points time dimension (pre-stimulation, stimulation and post-stimulation). Six nested mixed-effects models were tested. In each model, RMSSD score was the dependent variable (Table 5).

The likelihood ratio test showed that Model 3 was the best at predicting the collected data. The main effect of Type of touch and the interaction effect between Time and Type of touch emerged (Table 6) suggesting that newborns modulate their heart rate variability based on the type of tactile stimulation and that such modulation extends also to the resting period following the stimulation (Figure 4).

Finally, in order to explore individual differences in physiological responsiveness to tactile stimulation, we performed some correlational analysis between RMSSD scores and neonatal information (e.g., gestational age, birth weight, type of delivery and age at the moment of the test). First, we took into consideration the RMSSD values during the first resting period, which represent a baseline condition. The results revealed a significant correlation between hours after delivery and resting RMSSD (r = 0.39, *p* = 0.035). On the contrary, all the other parameters (gestational weeks, birth weight and type of delivery) did not correlate with baseline RMSSD (Figure 5).

Then, we took into consideration the differential RMSSD scores during the tactile stimulations, which was calculated by subtracting the RMSSD value measured during the pre-stimulus periods from the RMSSD value measured during the touch periods, both affective and non-affective. This score represents the physiological responsiveness to different tactile stimulations. The results revealed a positive correlation between the variation of RMSSD during affective touch and gestational age (r = 0.41, *p* = 0.027), but not with the other parameters. For the non-affective touch conditions, variation of RMSSD positive correlated with birth weight (r = 0.33, *p* = 0.077) and negative correlated with postnatal age (r = −0.32, *p* = 0.86), although statistical significance was not reached. The other two parameters (gestational weeks and type of delivery) did not correlate with variation of RMSSD during non-affective touch (Figure 6).

## 4. Discussion

In the present study we investigated newborns’ ability to discriminate between different types of tactile stimulation and to adaptively modulate their physiological state in response to the sensory input. One of the physiological mechanisms at the basis of self-regulation skills and arousal modulation seems to be mediated by parasympathetic responses (vagal tone), as indexed by modulation of HRV [21]. In the present study changes in HRV were measured as the variation of RMSSD score during tactile stimulation compared to pre-stimulus resting state. Tactile conditions differed for the spatio-temporal dynamics of the touch (stoking vs. tapping) as well as for the socio-affective meaning conveyed by source of touch (skin-to-skin contact vs. object contact). We predicted that gentle stroking performed by hand (affective touch) would promote vagal activity, as reflected by an increase in infants’ HRV, whereas tapping with a brush (non-affective touch) would be associated with vagal withdrawal, as reflected by a decrease in infants’ HRV.

The results are partially in line with our initial hypotheses as newborns’ showed physiological sensitivity to different types of tactile stimulation. This finding provides interesting evidence to support that newborns are able to differentiate between the two types of touch they were exposed to and to modulate their physiological state according to the specific type of touch experienced. Specifically, during the non-affective tactile stimulation, newborns displayed lower HRV, indexed by a decrease of RMSSD compared to the pre-stimulus resting period, indicating a vagal withdrawal. Thus, newborns showed physiological reactivity preparing the organism to process sensory information, which may reflect an unexpected change of surrounding environment. When affective states shift from calm to more activated states due to increased demands on infants sensory processing, vagal withdrawal can rapidly modulate heart rate to provide increases in cardiac output to support metabolically costly physiological responses to environmental stimulation [82]. On the contrary, during the affective touch stimulation, newborns did not show a modulation of HRV compared to the pre-stimulus resting period. This result is partially unexpected and needs some consideration. A previous study with one- to four- month-old infants found an increase of HRV during gentle stroking compared to static touch, pointing to an increased vagal activity mediated by the activation of the CT system [56], in line with evidence from tactile interventions on preterm infants, which suggest a beneficial effect of touch on behavioral and physiological regulation [44,45,83]. However there are some important methodological differences between the present study and the study of Van Puyvelde [56] that may account for the inconsistency of these results. First and most important, we tested newborns when they were in a sleeping state, resting supine in their cradle, instead of awake sitting on their mother’s lap. In newborns, autonomic activity changes during sleep states compared to waking state. Specifically, parasympathetic activity is dominant during quiet sleep as reflected by an increase in the high-frequency component of HRV [84]. Moreover, supine position modulates HRV, also enhancing high-frequency HRV [85]. Thus, in the present study newborns were tested in a condition that maximized the vagal influence on the heart, which may explain why affective touch did not result in a further HRV increase. We can speculate that the absence of HRV modulation when newborns were gentle stroked is link to the maintenance of a calm physiological state. Further investigation should explore the effect of affective touch when newborns are in wake active state or when they show distress, in order to better understand whether affective touch would promote autonomic regulation during a state of high arousal.

Another consideration is that in the present study infants were lying in their cradle without any physical contact other than the tactile stimulation manipulated by the experiment. Previous studies measured infants’ physiological responses to touch, while sitting on their mother’s lap [35,56]. Maternal presence represents a multisensory parenting system that combines a variety of information (e.g., vestibular, thermal and olfactory stimulation) that could have a soothing effect on infants’ behavior [86]. Indeed, it has been shown that simple physical contact can be effective in modulating infants’ cardio-respiratory rhythm according to the mother’s heartbeat, establishing a mother-infant physiological synchrony [87,88]. Therefore, it is possible that in previous studies the ecological context of maternal care contributed to the psychophysiological impact of affective touch, based on the integration of tactile stimulation with other social cues. Whereas, the experimental setting we designed focused on isolating the tactile stimulation in order to investigate the newborns’ ability to regulate autonomic responses to specific somatosensory input without any confounding effect related to other co-occurring sensory stimulations. In this consideration, the present results show for the first time that immediately after birth newborns are able to modulate their physiological state exclusively on the basis of somatosensory input, showing a very early sensitivity to tactile information.

Moreover, it’s important to point out that current results suggest that the effects of touch extend beyond the period of actual tactile stimulation, in line with previous studies that found a prolonged influence of touch on infants’ physiological state [45]. Importantly, there is evidence suggesting that infants who are able to modulate autonomic activity in response to environmental stimuli and to appropriately organize sensory information develop more efficient behavioral strategies for communication and social interactions [20]. Therefore, sensory regulatory capacity paves the way for the development of higher order regulatory processes, including emotional cognitive and behavioral regulation, which are progressively integrated and hierarchically organized in a sequential development of regulatory functions across the first years of life [3]. In the neonatal period, the main regulatory objective is related to homeostatic balance and organism-environmental exchange; then these needs gradually became part of a more extended aim of emotional regulation to cope with internal and external emotional demands and stress; in tune, regulatory processes extend to attention and goal-directed behavior, progressively including more sophisticated abilities, such as self-reflection [3]. Thus, tactile interactions may represent a fundamental component of physiological regulation in the neonatal period and a precursor for the emergence of socio-emotional and cognitive regulatory skills across development.

Furthermore, an important point of strength of the present study is that we used a within-subject design, as newborns experienced both tactile stimulations. Considering that HRV presents a huge interpersonal variability that originates during fetal development and endures consistently during postnatal life [89], it is essential that HRV analyses must be interpreted on an individual base, taking into consideration the changes on HRV in relation to individual baselines [65]. In order to better explore individual differences, we run some correlational analyses between HRV and neonatal information. Positive relation between RMSSD during baseline resting and postnatal age (hours after the delivery) emerged, in line with developmental studies that explored perinatal changes in HRV based on continuously ECG recording in the first days of life [65]. Shortly after birth, particularly during the first 6 h, HRV parameters change, showing an increase of HRV followed by stabilization, reflecting maturation of autonomic cardiovascular control [65]. Interestingly, postnatal age also correlated with newborns variation of HRV during non-affective touch, which may reflect an increasing responsiveness to sensory environment in the first days of life. While affective touch did not modulate HRV and such result was not affected by postnatal age, the decrease of HRV in response to non-affective touch showed a relation with postnatal age, suggesting an increasing ability to discriminate and differentiate physiological responses to different tactile stimulations. Moreover, the results revealed a relation between gestational age and the variation of RMSSD during affective touch, suggesting that infants born at lower gestational age showed a decrease of RMSSD during affective touch compared to the pre-stimulus resting period, whereas infants born at higher gestational age showed an increased HRV during affective touch. This result should be further explored in a large sample of newborns, including pre-term newborns to better understand the relation between gestational age and cardiac autonomic regulation in response to tactile stimulation. It is possible to speculate that at younger gestational age, newborns perceive any type of tactile contact as more arousing showing a vagal withdrawal. Indeed, it has been found that preterm newborns react in different ways to tactile stimulation: for very preterm newborns, who have a very fragile skin [90], touch may be stressful rather than soothing, while for newborns slightly premature it can be beneficial [45]. This suggests that touching has to be done with great care according to the newborns’ responses and needs.

Finally, the results of the present research allow for further consideration about the possible beneficial effect of external support, provided through affective touch, for infants who present scarce ability to adaptively cope with challenging sensory stimuli. By promoting the maintenance of an optimal physiological state, affective touch may enhance adaptation and self-regulation in response to internal and external environmental demands. This may be particular relevant for infants and children with sensory processing disorders, who show atypical responsiveness to sensory stimulation, lower autonomic self-regulation and inability to restore homeostasis after a stressor or a challenge, which influence their ability to adapt and interact in everyday activities [91,92]. It is hypothesized that these infants may have aberrant autonomic activity that underlies their sensory dysfunction, which results in been disturbed or distressed by typical levels of sensation, leading to possible difficulties in paying attention, risk for learning and behavioral difficulties and social isolation [3]. The present results contribute to the current knowledge with findings that suggest a possible application of affective touch in early sensory interventions that promote autonomic regulation. Future studies should specifically examine whether affective touch can be effective for helping infants and children with sensory difficulties to participate more successfully in their physical and social environment.

## 5. Conclusions

In conclusion, HRV reflects a dynamical cardiac regulation depending on the activity of the ANS in response to external stimulation. In the present study newborns show physiological sensitivity to different types of touch. Specifically, when newborns are in a calm state, non-affective touch elicits a vagal withdrawal as reflected by a decrease in HRV compared to the preceding resting period, whereas affective touch does not result in a change of HRV, possibly indicating a stable maintenance of physiological state. These results suggest that the ability to discriminate and to adaptively respond to different tactile sensory input via autonomic regulation develops very early in life with important implication for maintenance of physiological homeostasis and responsiveness to environmental stimulation. Thus, early tactile experiences provide a framework for infants’ ability to interact and assimilate information from the surrounding environment, critically shaping the development of socio-emotional and cognitive abilities.

## Figures and Tables

**Figure 1 ijerph-18-02212-f001:**
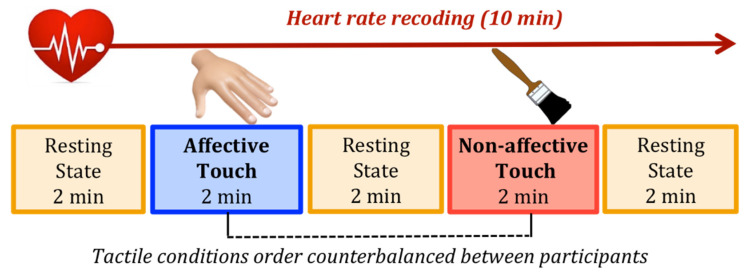
Example of experimental session: newborns’ heart rate was recorded for 10 min, alternating 2 min of resting with 2 min of tactile stimulation (affective touch vs. non-affective touch) in counterbalance order between participants.

**Figure 2 ijerph-18-02212-f002:**
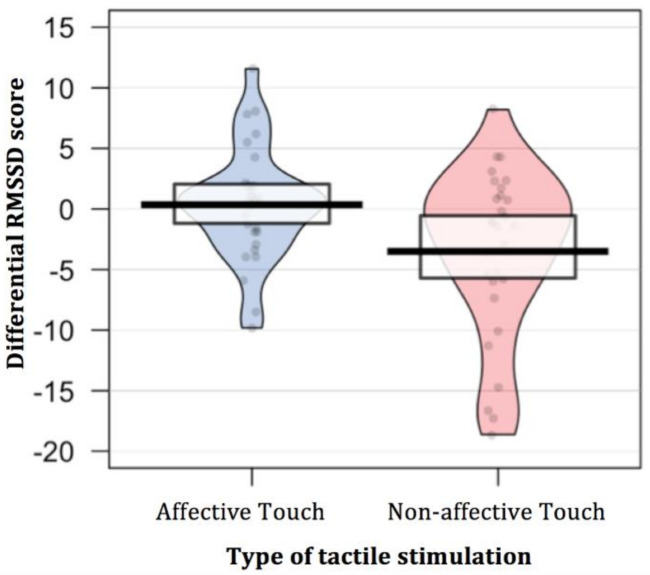
Differential RMSSD score during the two tactile stimulations compared to the pre-stimulus resting period. In the affective touch condition infants’ did not show a variation of RMSSD in response to touch, whereas in the non-affective touch condition they showed a decrease in RMSSD.

**Figure 3 ijerph-18-02212-f003:**
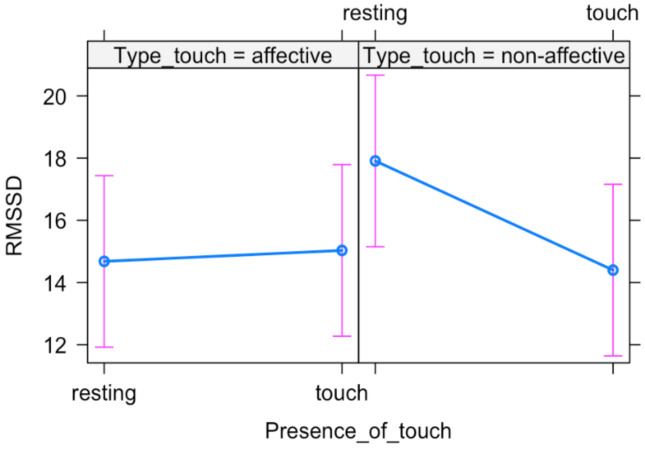
Plot of the interaction effect between Presence of touch and Type of touch. Mean (blue dots) and standard errors (pink lines) are displayed for both resting and tactile stimulation in each experimental condition (affective vs. non-affective).

**Figure 4 ijerph-18-02212-f004:**
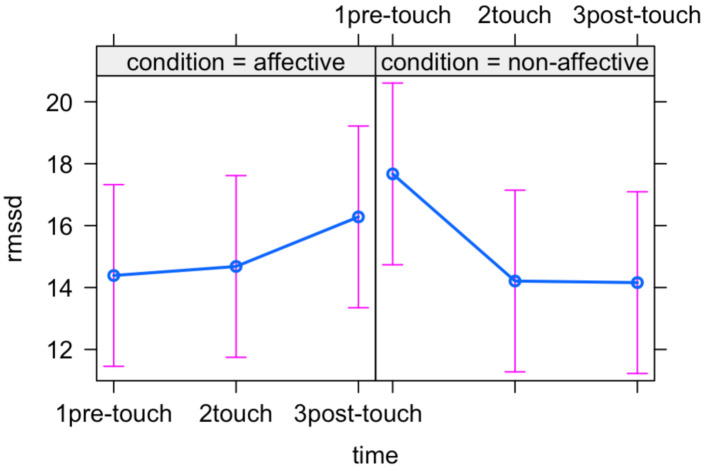
Plot of the interaction effect between Time and Type of touch. Mean (blue dots) and standard errors (pink lines) are displayed for each time point (pre-stimulation, stimulation and post-stimulation) in the two experimental conditions (affective and non-affective).

**Figure 5 ijerph-18-02212-f005:**
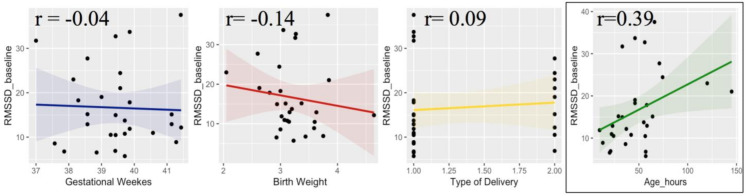
Correlation between RMSSD score during resting baseline and neonatal information, including gestational weeks (blue), birth weight (red), type of delivery (yellow) and postnatal age in hours (green).

**Figure 6 ijerph-18-02212-f006:**
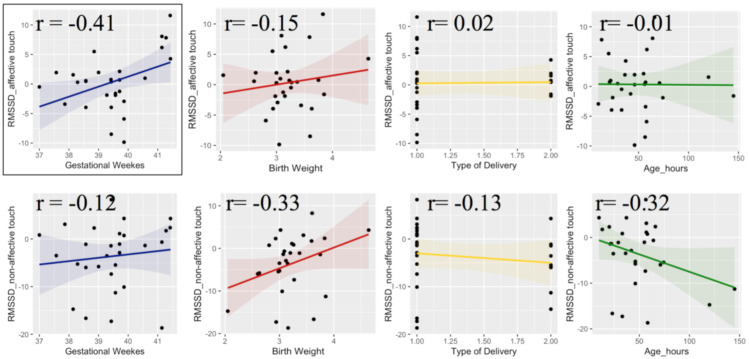
Correlation between RMSSD variation during tactile stimulations and neonatal information, including gestational weeks (blue), birth weight (red), type of delivery (yellow) and postnatal age in hours (green).

**Table 1 ijerph-18-02212-t001:** Descriptive statistics for each type of tactile stimulation (mean and standard deviation of the differential RMSSD score); simple *t*-test comparing the differential RMSSD score with the null level; and paired *t*-test comparing the differential RMSSD score in response to the two different tactile stimulations (affective touch vs. non-affective touch).

Type of Touch	Differential RMSSD Score	Simple *t*-Test (Null Level)	Paired *t*-Test
Affective Touch	0.35 (4.66)	t (29) = 0.41; *p* = 0.682 Cohen’s d = 0.07	t (29) = 2.71; *p* = 0.011 Cohen’s d = 0.66
Non-affective Touch	−3.51 (6.83)	t (29) = −2.81; *p* = 0.009 Cohen’s d = −0.51	

**Table 2 ijerph-18-02212-t002:** Descriptive statistics for each type of tactile stimulation (mean and standard deviation of the differential RMSSD score); simple *t*-test comparing the differential RMSSD score with the null level; and paired *t*-test comparing the differential RMSSD score in response to the two different tactile stimulations (affective touch vs. non-affective touch).

Type of Touch	Differential RMSSD Score	Simple *t*-Test (Null Level)	Paired *t*-Test
Affective Touch	1.90 (9.19)	t (29) = 1.13; *p* = 0.267 Cohen’s d = 0.21	t (29) = 2.04; *p* = 0.051 Cohen’s d = 0.62
Non-affective Touch	−3.47 (8.03)	t (29) = −2.37; *p* = 0.025 Cohen’s d = −0.43	

**Table 3 ijerph-18-02212-t003:** Comparison between models predicting RMSSD. Note that smaller values of AIC indicate better fitting models.

Tested Models	Variables	AIC	Delta AIC	Marginal R^2^	χ^2^	*p*
Model 0	Random effect of Participants	780.84				
Model 1	+ Presence of touch	779.49	2.88	0.011	3.349	0.067
Model 2	+ Type of touch	779.15	1.83	0.017	2.337	0.126
Model 3	+ Interaction Presence * Type of touch	775.76	6.13	0.033	5.393	0.020
Model 4	+ Order of tactile stimulation	777.06	0.15	0.035	0.704	0.401
Model 5	+ Interaction with Order	780.41	8.58	0.041	2.651	0.449

**Table 4 ijerph-18-02212-t004:** Summary of the most plausible-fitting model predicting RMSSD.

Variables	B (SE)	T	*p*
Presence of touch	0.352 (1.178)	0.299	0.766
Type of touch	3.229 (1.178)	2.741	0.007
Presence of touch × Type of touch	−3.861 (1.666)	−2.318	0.023

**Table 5 ijerph-18-02212-t005:** Comparison between models predicting RMSSD. Note that smaller values of AIC indicate better fitting models.

Tested Models	Variables	AIC	Delta AIC	Marginal R^2^	χ^2^	*p*
Model 0	Random effect of Participants	1173				
Model 1	+ Time	1173	1.99	0.006	2.935	0.231
Model 2	+ Type of touch	1175	−0.68	0.006	0.042	0.838
Model 3	+ Interaction Time × Type of touch	1170	10.95	0.024	9.400	0.009
Model 4	+ Order of tactile stimulation	1171	0.89	0.028	1.722	0.190
Model 5	+ Interaction with Order	1176	13.86	0.036	4.759	0.446

**Table 6 ijerph-18-02212-t006:** Summary of the most plausible-fitting model predicting RMSSD.

Variables	B (SE)	T	*p*
Time (PreTouch–Touch)	0.29 (1.24)	0.23	0.814
Time (PreTouch–PostTouch)	1.89 (1.24)	1.528	0.128
Type of touch Time (PreTouch–Touch) × (NonAffective) Time (PreTouch–PostTouch) × (NonAffective)	3.28 (1.24) −3.75 (1.75) −5.40 (1.75)	2.649 −2.142 −3.084	0.009 0.034 0.002

## Data Availability

The data presented in this study are available on request from the corresponding author. The data are not publicly available due to privacy restrictions.

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
