# Peer review of "In Touch with the Heartbeat: Newborns’ Cardiac Sensitivity to Affective and Non-Affective Touch"

_ijerph, 2021, doi:10.3390/ijerph18052212_

Round 1

Reviewer 1 Report

Dear authors,

please see the attached document for comments

Reviewer 2 Report

In this interesting work, the authors explore the relation between affective and non-affective touch and heart rate variability in newborns (N=30) while asleep or in a calm state. The focus and the findings of this work are presented and discussed in the context of autonomic self-regulatory processes and abilities.

  1. While I find the implications of this study of great interest, I have a methodological concern which may limit the interpretation of the authors as they are currently presented.

In fact, the two types of stimulations don’t only differ because one is an affective (skin-to-skin) touch and the other is non-affective (delivered with a paintbrush) but also the modality of the touch is different, i.e. stroking vs. tapping. What if the tapping was also delivered by a hand? Or if the gentle stroking is done with a brush? I believe this is an important point that needs discussion. Ideally, an additional experiment could be included as a control, but I’m aware that one strength of this study is that of adopting a within-subject design.

  1. The authors hypothesised that affective touch would promote vagal activity, as reflected by an increase in infants’ HRV, whereas non-affective touch would be associated with vagal withdrawal, as reflected by a decrease in infants’ HRV.

Overall, if I understood correctly, there are no significant effects of affective touch, i.e. there’s no modulation of HRV following affective touch. Therefore I don’t think it is valid to say that affective touch ‘promotes’ the maintenance of a stable HRV when no changes are observed, and the text should be clearer on this point.

  1. The significant effects are instead found in relation to a modulation of HRV following non-affective touch, i.e. a lower HRV. More explanation and discussion on such vagal withdrawal following non-affective touch is needed. I’d encourage the authors to further expand and explain the sentence (line 401-402) “newborns showed physiological reactivity preparing the organism to face sensory challenge in response to a non-affective tactile stimulation.”

  1. The authors also looked at the effect of touch post-stimulus in the following 2 minutes. The results on this section are not clear. The text which refers to Table 4 appears to be incorrect, and Table 4 is to be edited with the variables and the values clearly aligned.

  1. The stats reported at line 325 also seem incorrect and do not match those reported in Table 5. “On the contrary, when infants were stimulated with affective touch, they did not show a modulation of HRV compared to the pre-stimulus resting period [t (29)=0.41, p= .011, Cohen’s d= 0.07; p-value adjusted for multiple comparisons using Bonferroni correction, p< 0.025] (Table 5; Figure 4).”

  1. More generally, while I believe that all the hypotheses and analyses included are valid and sound, I think they can be better and more clearly organised. I’d encourage the authors to improve clarity by numbering the hypotheses and systematically matching the planned statistical analyses and the results.

  1. I’d also recommend to introduce the RMSSD index in the introduction section, and expand a bit on what this index reflects as it then becomes the main measure of the paper.

  1. Finally, there are a typos and grammar errors throughout that need to be checked and corrected.

Reviewer 3 Report

In the present study, the authors investigated the effect of touch on newborns and measured somatosensory stimulations which are indicated by modulation of heart rate variability (HRV). Two minutes of affective (stroking) and non-affective (tapping) touch on the newborn's forehead were applied and HRV was measured. Between each touch, there were 2 minutes of resting. Their finding revealed that newborns showed a positive response towards affective touch than non-affective touch. Newborns showed stable HRV when they experienced affective touch. Overall this study provides the effect of touch as early somatosensory stimulation on autonomic self-regulation, social and cognitive development in newborns.

Here are my comments/suggestion to improve the study:

  1. Describe more about Apgar score. There was no information was provided about the Apgar score in the methods section.
  2. Please write  the rationale behind selecting 2 min interval.
  3. Was there a significant difference between heart rate during affective touch and pre resting state? Also, was there any significant difference between heart rate during affective touch and post resting state?
  4. The authors should describe more about statistical analysis in the methods section. Authors have used a mixed model and it would be nice if they provide an interaction model in the methods section.

Round 2

Reviewer 1 Report

Thank you for considering all the comments and for improving the paper. I do not have any further suggestions.